# Efficient Text-driven Human Motion Generation via Latent Consistency Training

## Abstract

Consistency models excel at few-step inference in generative tasks across various scenarios, but typically rely on pre-trained diffusion model distillation, involving additional training costs and performance limitations. In this paper, we propose a *motion latent consistency training* framework that learns directly from data rather than distillation for efficient and text-controllable human motion generation. For representation optimization, we design a motion autoencoder with quantization constraints that enable concise and bounded motion latent representations. Focusing on conditional generation, we construct a *classifier-free guidance* (CFG) format with an additional unconditional loss function that extends the CFG technique from the inference phase to the training phase for conditionally guided consistency training. We further propose a clustering guidance module to provide additional references to the solution distribution at minimal query cost. By combining these enhancements, we achieve stable and consistent training in non-pixel modality and latent representation spaces for the first time. Experiments in benchmarks demonstrate that our method significantly outperforms traditional consistency distillation methods with reduced training cost, and enhances the consistency model to perform comparably to state-of-the-art models with lower inference cost. Our code will be open source.

## 1 Introduction

Synthesizing human motion sequences from specific text prompts is a fundamental task in robotics and virtual reality. Recent advancements in text-to-motion diffusion frameworks (Tevet et al., 2023; Zhang et al., 2022) have generated increasingly realistic and diverse motion sequences. These works (Chen et al., 2023; Kong et al., 2023; Jin et al., 2023; Lu et al., 2022a) exhibit powerful distribution estimation capabilities and controllability, but at the cost of a hundred-fold increase in computational burden involved in the expensive and numerous function evaluation iterations required. For efficient sampling, previous work (Chen et al., 2023; Kong et al., 2023) has attempted to introduce numerical solvers (Liu et al., 2022) to solve rapidly within well-designed latent spaces. However, larger sampling strides are associated with large numerical errors due to the nonlinear nature of the diffusion trajectories, causing significantly reduced fidelity of these methods at lower NFEs. Efficiency bottlenecks in motion diffusion frameworks is emerging as a critical bottleneck in its application.

Recent advances attempt to shift expensive iterations to the training phase and learn pre-computed diffusion trajectories for large-scale skip-step or single-step sampling during inference, which are known as the *consistency model* (Song et al., 2023). Typical precalculated trajectory methods are *consistency distillation* (Luo et al., 2023; Wang et al., 2023) and *consistency training* (Yang & Prafulla, 2024; Kong et al., 2024). Consistency distillation rely on a well-trained diffusion model as the teacher, and training them from scratch is both computationally expensive and time-consuming. Additionally, the distillation process is constrained by the sample quality of the teacher model, which caps the performance ceiling. Conversely, consistency training with lower training costs, which calculate the log probability gradient $\nabla_{x_t} \log p(x_t)$ directly from raw data during the reverse diffusion stage, avoid these limitations. However, estimating trajectory distributions from individual raw data presents greater challenges than distillation guidance, resulting in suboptimal performance. Despite the advances (Yang & Prafulla, 2024) in raw pixel representation in recent work, such performance challenges of consistency training in non-pixel modalities, especially latent spaces, remain serious.

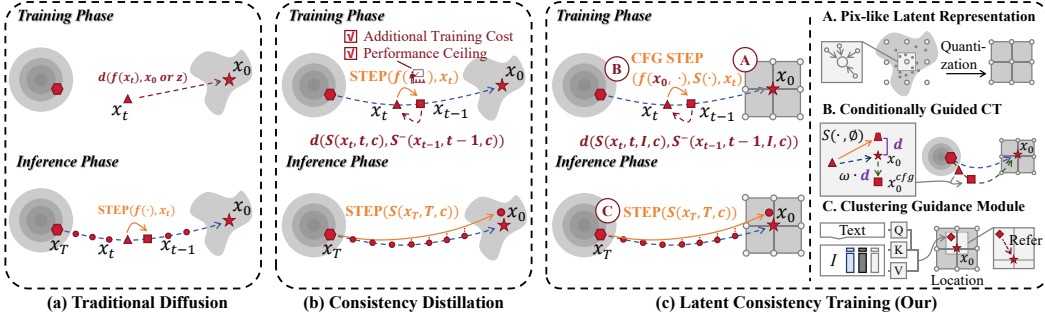

Figure 1: **Overview of the distinctions between our method with traditional methods.** (a) Traditional diffusion methods calculate diffusion trajectories at the inference phase, involving expensive sampling iteration costs. (b) Consistent distillation precalculates the diffusion trajectories in the training phase through the teacher model and constrains the metric loss from the output of the consistency model $S$ between adjacent perturbed states to achieve few-step sampling in inference. (c) Consistent training escapes the constraints of the pre-trained model. Additionally, we optimize the latent representation as bounded and concise and present the conditional guidance and clustering guidance to optimize the diffusion trajectories from individual raw data.

To tackle these challenges, we propose the *motion latent consistency training* (MLCT) framework from the following three aspects. (i) **Latent representation design.** Variational motion representations based on Kullback-Leibler (KL) constraints struggle in consistency training, since precisely inferring diffusion trajectories in unbounded continuous solution spaces is intractable without teacher guidance. Inspired by the success of consistency training in pixel space, our first insight is to extend a motion autoencoder with the quantization constraint to construct pix-like latent representations with bounded and finite states. To this end, we restrict the representation boundaries with the hyperbolic tangent (Tanh) function and force the continuous representation to map to the nearest predefined clustering center. Such representations offer simplified solution spaces and quantization mechanism between adjacent state contributes to counteracting cumulative errors in consistency model. In addition, previous practice (Lu et al., 2022b) demonstrates that the boundedness of the representations contributes to sustaining stable inference in classifier-free guidance (CFG) techniques (Ho & Salimans, 2021). (ii) **Conditional guidance.** Traditional consistency training neglects conditional trajectory guidance since the latter is essentially enhancement techniques for the inference phase of diffusion models and relies heavily on well-trained diffusion models as preconditions. Our second insight is to present a conditionally guided consistency training framework based on CFG format online simulation. It treats the ground truth latent representation as the simulation of the conditional prediction and replaces the unconditional estimation with an online updated model based on the additional loss term. The constructed CFG format facilitates distinguishing diffusion trajectories across conditions in highly perturbed states. (iii) **Clustering guidance.** For traditional consistency models, the current perturbed state solution distribution is guided only by the previous perturbed state, resulting in an inefficient training process. Our third insight is to propose a clustering guidance module based on the attention-like calculation and the K-Nearest Neighbor (KNN) algorithm. Specifically, we utilize KNN to construct clustering dictionaries with textual representation cluster centers as keys and mean motion representations values in the same category as values. It leverages an attention-like query mechanism to provide solution distribution references based on given textual conditions.

Our contributions are four-fold: (1) We extend motion latent representations based on the quantization constraint, which are bounded finite states, providing a powerful latent space embedding scheme in the consistency training framework. (2) We present the conditionally guided consistency training framework, which extends CFG from the inference phase to the training phase. To the best of our knowledge, we have explored consistency training in latent space *for the first time*, and are also *the first* to introduce CFG into consistency training. (3) We propose a clustering guidance module that contributes to providing additional solution distribution references at minimal query cost. (4) Our work achieves performance matching state-of-the-art methods on two datasets: KIT and HumanML, with an inference speed of only 54 ms and without any diffusion model pre-training cost. Extensive experiments indicate the effectiveness of the proposed methods and each component.

## 2 RELATED WORK

**Human motion generation.** Human motion generation aims to synthesize human motion sequence under specified conditions, such as action categories (Lee et al., 2023; Xu et al., 2023), audio (Li et al., 2022; Pang et al., 2023), and textual description (Ahuja & Morency, 2019; Tevet et al., 2023; Chen et al., 2023). Recent advancements in multi-step generative methods have proven successful, notably auto-regressive (Zhang et al., 2023a; Guo et al., 2024) and diffusion methods (Zhang et al., 2022; Tevet et al., 2023; Chen et al., 2023). Among these, diffusion methods are increasingly prominent due to their stable distribution estimation and high-quality sampling results. Motiondiffuse (Zhang et al., 2022) and MDM (Tevet et al., 2023) were the pioneers in implementing diffusion frameworks for motion generation. MLD (Chen et al., 2023) realizes the latent space diffusion, which significantly improves the efficiency. ReMoDiffuse (Zhang et al., 2023b) explores initial state guidance through hybrid retrieval to generate more realistic motion. GraphMotion (Jin et al., 2023) leverages semantic role processing tools for fine-grain controllable generation. While these advances offer significant improvements in generation performance, the tradeoff is the high cost of sampling iterations. Efficient sampling strategies are categorized into two approaches: large-stride numerical sampling, which employs higher-order ordinary differential equation (ODE) approximation methods (Lu et al., 2022a; Song et al., 2021a), and pre-calculated diffusion trajectories, which are represented by diffusion distillation (Liu et al., 2022; Xu et al., 2022; Song et al., 2023). Numerical methods face constraints due to the nonlinear nature of diffusion trajectories, often requiring more than 20 function evaluations (NFE) to diminish numerical errors from large strides. Distillation methods use a well-trained diffusion model as a teacher to generate precomputed trajectories, enabling few-step generation, but incur high training costs and face performance ceilings. Recent advances include consistency models (Song et al., 2023), particularly the consistency training free from the distillation mode, which show promise for high-quality few-step generation at lower costs.

**Consistency model.** Consistency models (Song et al., 2023) are categorized into consistency distillation and consistency training based on precomputed trajectory methods. It achieves efficient trajectory distillation and single-step inference through maintaining consistency of model outputs on the same diffusion trajectory. Consistency distillation is a typical diffusion distillation method that relies on strong teacher model guidance and is adapted to well established diffusion model improvement techniques, such as CFG, Lora, and control net. The stable diffusion guidance enables its extension into various fields (Luo et al., 2023; Wang et al., 2023; Kim et al., 2023; Ye et al., 2023; Lu et al., 2024; Fei et al., 2024; Xiao et al., 2023). We note contemporaneous work (Dai et al., 2024) that extends consistent distillation to human motion generation tasks. However, constrained from the high training cost and performance ceiling of the teacher distillation mode, the existing methods remain significant gaps with the state-of-the-art diffusion frameworks. Comparatively, consistency training simulates diffusion trajectories within the raw data, freeing from the limitations of pre-trained models. Nevertheless, its performance is significantly inferior to distillation-based methods due to the lack of guidance, and the advances are stuck in the earliest proposed raw pixel representations. ICM (Yang & Prafulla, 2024) further explores and improves consistency training methods to obtain similar performance to consistency distillation without pre-trained models. But its research object is still raw pixel representations, and consistency training in non-image modal data as well as latent representations remains unexplored. Additionally, previous studies have neglected guidance techniques in consistency training, such as conditional guidance and initial state guidance, which offer potential for improvement. To address these shortcomings, our work focuses on constructing the latent consistency training paradigm to improve the performance of consistency models in motion modalities to state-of-the-art levels with lower training and inference costs.

## 3 PRELIMINARIES

### 3.1 SCORE-BASED DIFFUSION MODELS

The diffusion models (Ho et al., 2020) is a class of generative model that gradually injects Gaussian noise into the data and then generates samples from the noise through a reverse denoising process. Specifically, it gradually transforms the data distribution $p_{data}(x_0)$ into a well-sampled prior distribution $p(x_T)$ via a Gaussian perturbation kernel $p(x_t|x_0) = \mathcal{N}(x_t|\alpha_t x_0, \sigma_t^2 I)$, where $\alpha_t$ and $\sigma_t$ are noise schedules. Recent studies have formalized it into a continuous time form, described as

stochastic differential equations (SDEs),

$$dx_t = f(t)x_t dt + g(t)dw_t, \tag{1}$$

where $t \in [\epsilon, T]$, $\epsilon$ and $T$ are the fixed positive constant, $w_t$ denotes the standard Brownian motion, $f$ and $g$ are the drift and diffusion coefficients respectively. They are related the noise schedules as follows.

$$f(t) = \frac{d \log \alpha_t}{dt}, \quad g^2(t) = \frac{d\sigma_t^2}{dt} - 2\frac{d \log \alpha_t}{dt}\sigma_t^2. \tag{2}$$

Previous work has revealed that the reverse process of Equation 1 shares the same marginal probabilities with the *probabilistic flow ODE*:

$$dx_t = [f(t)x_t - \frac{1}{2}g^2(t)\nabla_{x_t} \log p(x_t)]dt, \tag{3}$$

where $\nabla_x \log p(x_t)$ is named the *score function*, which is the only unknown term in the sampling pipeline. An effective approach is training a time-dependent score network $\mathcal{S}_\theta(x_t, t)$ to estimate $\nabla_x \log p(x_t)$ based on conditional score matching, parameterized as the prediction of noise or initial value in forward diffusion. Further, Equation 3 can be solved in finite steps by numerical ODE solvers such as Euler (Song et al., 2021b) and Heun solvers (Karras et al., 2022). Upon the above study, previous work also has explored conditional probabilities $p(x_t|y)$ for the more controlled generation, where $y$ is the condition such as text or action. One successful approach is known as Classifier-Free Guidance (CFG), which is parameterized as a linear combination of unconditional and conditional noise predictions, i.e. $\tilde{z}_\theta(x_t, t, c) = (1 + \omega)z_\theta(x_t, t, c) - \omega z_\theta(x_t, t, \emptyset)$, where $\omega$ is guidance scale.

## 3.2 Consistency Models

Theoretically, the reverse process expressed by Equation 3 is deterministic, and the consistency model (Song et al., 2023) achieves one-step or few-step generation by pulling in outputs on the same ODE trajectory. It is formally expressed as,

$$\mathcal{S}_\psi(x_t, t) \approx \mathcal{S}_\psi(x_{t'}, t') \approx \mathcal{S}_\psi(x_\epsilon, \epsilon) \approx \epsilon \quad \forall t, t' \in [\epsilon, T], \tag{4}$$

which is known as *the self-consistency property*. To maintain the boundary conditions, existing consistency models are commonly parameterized by skip connections, i.e.,

$$\mathcal{S}_\psi(x_t, t) := c_{skip}(t)x_t + c_{out}(t)\hat{\mathcal{S}}_\psi(x_t, t) \tag{5}$$

where $c_{skip}(t)$ and $c_{out}(t)$ are differentiable functions satisfied $c_{skip}(\epsilon) = 1$ and $c_{out}(\epsilon) = 0$. For stabilize training, the consistency model maintaining target model $\mathcal{S}_\psi^-$, trained with the exponential moving average (EMA) of rate $\gamma$, that is $\psi^- \leftarrow \gamma\psi^- + (1 - \gamma)\psi$. The consistency loss can be formulated as,

$$\mathcal{L}_{cm} = \mathbb{E}_{x,t}\big[d\big(\mathcal{S}_\psi(x_{t_{n+1}}, t_{n+1}), \mathcal{S}_{\psi^-}(\hat{x}_{t_n}, t_n)\big)\big] \tag{6}$$

where $d(\cdot, \cdot)$ is a metric function such as mean square or pseudo-huber metric, and $\hat{x}_{t_n}$ is a one-step estimation from $x_{t_{n+1}}$ with ODE solvers applied in Equation 3.

As indicated in Equation 6, the output of the current state in the traditional consistency model is exclusively determined by the output of the preceding state. It leads to cumulative errors between the outputs of adjacent perturbed states being transferred to the initial state along the diffusion trajectory, and lacks immediate reference to the solution distribution.

## 4 Method

In this section, we construct the latent consistency training paradigm from three perspectives: latent representation design, conditional guidance, and clustering guidance, as illustrated in Figure 2.

### 4.1 Latent Representation with the Quantization Constraint

For motion representation optimization, we construct the motion autoencoder $\mathcal{G} = \{\mathcal{E}, \mathcal{D}\}$ for encoding and reconstruction between the raw motion sequence $x$ and the motion latent representation

Figure 2: **Approach overview.** (a) Motion sequences are encoded through quantization constraints (QC) and characterized as bounded finite states, analogous to pixel representations. (b) Constructing conditional diffusion trajectories during the training phase via CFG format online simulation. (c) Providing the solution distribution reference based on the given textual condition via constructing the clustering dictionary using the KNN algorithm and employing an attention-like query mechanism.

$z$. It relies on quantization constraints to ensure boundedness and regularity of $z$. Specifically, each dimension of $z$ is sampled from a finite set $\mathcal{M}$ of size $2l + 1$ as follow,

$$\mathcal{M} = \{z_i; -1, -j/l, \cdots, 0, \cdots, j/l, \cdots, 1\}_{j=0}^{l}. \tag{7}$$

For brevity, we denote $l$ as the quantization level. It is structurally analogous to the normalized primitive pixel representation and shares the distinctive characteristics of finite continuous states and enforced intervals between adjacent states. Our work denotes $z \in \mathbb{R}^{n,d}$ as $n$ learnable tokens with $d$ dimension, aggregating the motion sequence features via attention computation (Vaswani et al., 2017). The hyperbolic tangent (*Tanh*) function is employed on the output of the encoder $\mathcal{E}$ to constrain the boundaries of the representation, and then quantize the result by the round operator $\mathcal{R}$. Furthermore, the gradient of quantized items is simulated by the previous state gradient to back-propagate the gradient normally, which is known as the straight-through estimator (STE) (Bengio et al., 2013). The latent representations $z_m$ are sampled by the following format,

$$z_m = \mathcal{R}\Big(l \cdot tanh(\mathcal{E}(x))\Big)/l. \tag{8}$$

The proposed approach diverges from earlier quantitative research (Mentzer et al., 2023) by emphasizing the robustness of the continuous representation generated through forced clustering via quantization constraints, whereas prior studies primarily concentrate on the discrete characteristics of quantization for codebook construction. Due to memory and computational limitations associated with codebooks, previous work often employs a limited number of candidate states, which constrains reconstruction performance. In contrast, our method does not rely on a codebook, enabling a greater number of candidate states to be incorporated into $\mathcal{M}$, thereby reducing reconstruction error.

The standard optimization target is to reconstruct motion information from $z$ with the decoder $\mathcal{D}$, i.e., to optimize the $l_1$ smooth error loss,

$$\mathcal{L}_z = \mathbb{E}_x\Big[d\Big(x, \mathcal{D}(z_m)\Big) + \lambda_j d\Big(\mathcal{J}(x), \mathcal{J}(\mathcal{D}(z_m))\Big)\Big], \tag{9}$$

where $\mathcal{J}$ is a function to transform features such as joint rotations into joint coordinates. $\lambda_j$ is the balancing weight.

## 4.2 Conditionally Guided Consistency Training

The diffusion stage begins with the variance preserving schedule (Song et al., 2021b) to perturbed motion latent representations $x_\epsilon = z$ with perturbation kernel $\mathcal{N}(x_t; \alpha(t)x_0, \sigma^2(t)I)$,

$$\alpha_t := e^{-\frac{1}{4}t^2(\beta_1 - \beta_0) - \frac{1}{2}t\beta_0}, \quad \sigma_t := \sqrt{1 - e^{2\alpha(t)}}. \tag{10}$$

The consistency model $\mathcal{S}_\theta$ has been constructed to predict $x_\epsilon$ from perturbed $x_t$ in a given PF-ODE trajectory. To maintain the boundary conditions that $\mathcal{S}_\psi(x_\epsilon, \epsilon, c) = x_\epsilon$, we employ the same skip

setting for Equation 5 as in LCM (Luo et al., 2023), which parameterized as follow:

$$\mathcal{S}_\psi(x_t, t, c) := \frac{\eta^2}{(10t)^2 + \eta^2} \cdot x_t + \frac{10t}{\sqrt{(10t)^2 + \eta^2}} \cdot \widetilde{\mathcal{S}}_\psi(x_t, t, c), \tag{11}$$

where $\widetilde{\mathcal{S}}_\psi$ is a transformer-based network and $\eta$ is a hyperparameter, which is usually set to 0.5. Following *the self-consistency property* (as detailed in Equation 4), the model $\mathcal{S}_\psi$ has to maintain the consistency of the output at the given perturbed state $x_t$ with the previous state $\widetilde{x}_{t-\Delta t}$ on the same ODE trajectory. The latter can be estimated from Equation 3 via the DPM++ solver:

$$\widetilde{x}_{t-\Delta t} \approx \Phi(x_\epsilon^\Phi, x_t, x_{t-\Delta t}) = \frac{\sigma_{t-\Delta t}}{\sigma_t} \cdot x_t - \alpha_{t-\Delta t} \cdot (e^{-h_t} - 1) \cdot x_\epsilon^\Phi, \tag{12}$$

where $h_t := \lambda_{t-\Delta t} - \lambda_t$, $\lambda_t := \log(\alpha_t/\sigma_t)$, and $x_\epsilon^\Phi$ is the estimation of $x_\epsilon$ under the different sampling strategies. In particular, $x_\epsilon^\Phi$ can be parameterized as a linear combination of conditional and unconditional latent presentation prediction following the CFG strategy, i.e.,

$$x_\epsilon^\Phi(x_t, t, c) = (1 + \omega) \cdot \mathcal{F}_\psi(x_t, t, c) - \omega \mathcal{F}_\psi(x_t, t, \emptyset), \tag{13}$$

where $\mathcal{F}_\psi(\cdot)$ is well-trained and $x_\epsilon$-prediction-based motion diffusion model.

It is worth noting that $x_\epsilon$ can be utilized to simulate $\mathcal{F}_\psi(x_t, t, c)$ as used in the vanilla consistency training pipeline. Furthermore, $\mathcal{F}_\psi(x_t, t, \emptyset)$ can be replaced by $\mathcal{S}_\psi(x_t, t, \emptyset)$ with online updating based on the additional unconditional loss item. Thus Equation 13 can be rewritten as:

$$x_\epsilon^\Phi(x_t, t, c) = (1 + \omega) \cdot x_\epsilon - \omega \mathcal{S}_\psi(x_t, t, \emptyset). \tag{14}$$

We refer to Equation 14 as the *conditional trajectory simulation*. The optimization objective of the consistency model $\mathcal{S}_\theta$ is that,

$$\mathcal{L}_c = \mathbb{E}_{x,t}\Big[ \underbrace{\frac{1}{\Delta t}d\Big(\mathcal{S}_\psi(x_t, t, c), \mathcal{S}_{\psi^-}(\widetilde{x}_{t-\Delta t}, t - \Delta t, c)\Big)}_{\text{Consistency Loss}} + \underbrace{d\Big(\mathcal{S}_\psi(x_t, t, \emptyset), x_\epsilon\Big)}_{\text{Unconditional Loss}} \Big], \tag{15}$$

where $d(x, y) = \sqrt{(x - y)^2 + \gamma^2} - \gamma$ is pseudo-huber metric, $\gamma$ is a constant. The target network $\mathcal{S}_{\psi^-}$ is updated after each iteration via EMA. More details of consistency training setting as well as training and inference pseudo-code are shown in the Appendix B.

## 4.3 CLUSTERING GUIDANCE MODULE

To enhance solution distribution guidance under specific textual conditions, we present the clustering guidance module. Prior to training, a clustering dictionary is constructed for the training set. Specifically, we employ the K-Nearest Neighbor (KNN) algorithm to classify the embedded features of each text in the training set into $K$ classes. The clustering centers for each class are utilized as keys to construct the clustering dictionary, denoted as $\mathcal{K} \in \mathbb{R}^{K, d_c}$, where $d_c$ represents the dimension of the text representations. Subsequently, we calculate the mean values of the corresponding motion representations within the same text categories to establish the values of the clustering dictionary, denoted as $\mathcal{V} \in \mathbb{R}^{K, n, d_m}$ represent the token count and the dimension of the motion representations.

In the training and inference phases, the clustering dictionary is invoked via an attention-like computation. For instance, given a text instruction for constructing a query vector, denoted as $\mathcal{Q} \in \mathbb{R}^{1, d_c}$. The motion clustering guidance representation $\mathcal{I} \in \mathbb{R}^{1, n, d_m}$ can be computed in the following form:

$$\mathcal{I} = softmax(\mathcal{Q} \cdot \mathcal{K}^T) \cdot \mathcal{V}. \tag{16}$$

The clustering guidance provides a more flexible scheme that allows the model to rapidly localize the solution distribution at a lower query cost. To manage computational complexity, the query computation is performed only once during a single inference process. For the input $x^{(i)}$ of the $i$-th block in the backbone network, we map the query results $\mathcal{I}$ into dimensions consistent with the $x^{(i)}$ via a linear layer and implement feature fusion using an element-wise summation operator.

Table 1: Average inference time for single sample inference. It is first measured on the RTX 4090 GPU, and then aligned on the Tesla V100 GPU using the MLD as an intermediary benchmark.

| Method | MDM (Tevet et al., 2023) | MotionDiffuse (Zhang et al., 2022) | MLD (Chen et al., 2023) | GraphMotion (Jin et al., 2023) | ReMoDiffuse (Zhang et al., 2023b) | T2M-GPT (Zhang et al., 2023a) |
|---|---|---|---|---|---|---|
| AITS (s) | 24.74 | 14.74 | 0.217 | 1.495 | 0.417 | 0.598 |
| Method | AttT2M (Zhong et al., 2023) | MoMask (Guo et al., 2024) | MotionLCM (NFE 1) (Dai et al., 2024) | Our NFE 1 | Our NFE 2 | Our NFE 4 |
| AITS (s) | 0.717 | 0.118 | 0.030 | 0.031 | 0.038 | 0.054 |

Table 2: Comparisons to state-of-the-art methods on the HumanML3D test set. "↑" denotes that higher is better. "↓" denotes that lower is better. "→" denotes that results are better if the metric is closer to the real motion. The gray background indicates the sota method of the current framework. **Bold** and underlined indicate the best and second-best results, respectively.

| Method | R-Precision ↑ | | | FID ↓ | MM-Dist ↓ | Diversity → | MModality ↑ |
|---|---|---|---|---|---|---|---|
| | Top-1 | Top-2 | Top-3 | | | | |
| Real | $0.511^{\pm.003}$ | $0.703^{\pm.003}$ | $0.797^{\pm.002}$ | $0.002^{\pm.000}$ | $2.974^{\pm.008}$ | $9.503^{\pm.065}$ | - |
| MDM | $0.320^{\pm.005}$ | $0.498^{\pm.004}$ | $0.611^{\pm.007}$ | $0.544^{\pm.044}$ | $5.566^{\pm.027}$ | $\underline{9.559}^{\pm.086}$ | $\mathbf{2.799}^{\pm.072}$ |
| MotionDiffuse | $0.491^{\pm.001}$ | $0.681^{\pm.001}$ | $0.782^{\pm.001}$ | $0.630^{\pm.001}$ | $3.113^{\pm.001}$ | $9.410^{\pm.049}$ | $1.553^{\pm.042}$ |
| MLD | $0.481^{\pm.003}$ | $0.673^{\pm.003}$ | $0.772^{\pm.002}$ | $0.473^{\pm.013}$ | $3.196^{\pm.010}$ | $9.724^{\pm.082}$ | $2.413^{\pm.079}$ |
| GraphMotion | $0.504^{\pm.003}$ | $0.699^{\pm.002}$ | $0.785^{\pm.002}$ | $0.116^{\pm.007}$ | $3.070^{\pm.008}$ | $9.692^{\pm.067}$ | $2.766^{\pm.096}$ |
| ReMoDiffuse | $0.510^{\pm.005}$ | $0.698^{\pm.006}$ | $0.795^{\pm.004}$ | $0.103^{\pm.004}$ | $2.974^{\pm.016}$ | $9.018^{\pm.075}$ | $1.795^{\pm.043}$ |
| T2M-GPT | $0.491^{\pm.003}$ | $0.680^{\pm.003}$ | $0.775^{\pm.002}$ | $0.116^{\pm.004}$ | $3.118^{\pm.011}$ | $9.761^{\pm.081}$ | $1.856^{\pm.011}$ |
| AttT2M | $0.499^{\pm.003}$ | $0.690^{\pm.002}$ | $0.786^{\pm.002}$ | $0.112^{\pm.006}$ | $3.038^{\pm.007}$ | $9.700^{\pm.090}$ | $2.452^{\pm.051}$ |
| MoMask | $0.521^{\pm.002}$ | $0.713^{\pm.002}$ | $0.807^{\pm.002}$ | $0.045^{\pm.002}$ | $2.958^{\pm.008}$ | - | $1.241^{\pm.040}$ |
| Our (NFE 1) | $0.530^{\pm.002}$ | $0.726^{\pm.002}$ | $0.822^{\pm.002}$ | $0.264^{\pm.007}$ | $2.888^{\pm.007}$ | $9.799^{\pm.061}$ | $2.188^{\pm.049}$ |
| Our (NFE 2) | $\mathbf{0.538}^{\pm.003}$ | $\mathbf{0.734}^{\pm.002}$ | $\mathbf{0.828}^{\pm.002}$ | $\underline{0.094}^{\pm.003}$ | $\underline{2.822}^{\pm.005}$ | $9.595^{\pm.075}$ | $2.325^{\pm.061}$ |
| Our (NFE 4) | $\underline{0.537}^{\pm.003}$ | $\underline{0.732}^{\pm.002}$ | $\underline{0.826}^{\pm.002}$ | $\mathbf{0.060}^{\pm.003}$ | $\mathbf{2.819}^{\pm.010}$ | $9.545^{\pm.068}$ | $2.571^{\pm.051}$ |

## 5 EXPERIMENTS

### 5.1 EXPERIMENTAL SETUP

**Datasets.** We evaluate our framework on two mainstream benchmarks for text-driven motion generation tasks, which are the **KIT** (Plappert et al., 2016) and the **HumanML3D** (Guo et al., 2022). The former contains 3,911 motions and their corresponding 6,363 natural language descriptions. The latter is a large 3D human motion dataset comprising the HumanAct12 (Guo et al., 2020) and AMASS (Mahmood et al., 2019) datasets, containing 14,616 motions and 44,970 descriptions.

**Evaluation Metrics.** Consistent with prior research, we evaluate the proposed framework across four aspects. Motion Quality: we use the **Fréchet Inception Distance (FID)** to assess the distance between feature distributions of generated and real data. Condition Matching: we apply **R-Precision** to measure the correlation between text descriptions and generated motion sequences, recording the probabilities of the first matches for $k = 1, 2, 3$. We then calculate the distance between motions and texts using the **Multi-Modal Distance (MM Dist)**. Diversity: we assess feature differences with the **Diversity** metric and measure generative diversity for the same text input using the **Multi-modality (MM)** metric. Computational Burden: we measure inference efficiency using the **Average Inference Time per Sentence (AITS)** in seconds. Detailed metrics are shown in the Appendix E.

**Implementation Details.** The architecture of our network is consistent with the baseline model MLD (Chen et al., 2023). Specifically, both the encoder $\mathcal{E}$ and decoder $\mathcal{D}$ contain 7 layers of transformer blocks with input dimensions 256, and each block contains 4 learnable tokens. The quantization levels default set $l = 256$. The consistency model $\mathcal{S}$ contains 9 layers of transformer blocks with input dimensions 512. More training details are shown in the Appendix B.

### 5.2 COMPARISONS TO STATE-OF-THE-ART METHODS

To illustrate the efficiency advantage of our method, we present the average sampling time of the proposed approach in comparison to state-of-the-art methods, as shown in Table 1. Additionally,

Table 3: Comparisons to state-of-the-art methods on the KIT test set. Marker meaning is consistent with Table 2.

| Method | R-Precision ↑ | | | FID ↓ | MM-Dist ↓ | Diversity → | MModality ↑ |
|--------|-------|-------|-------|-------|-----------|-------------|-------------|
| | Top-1 | Top-2 | Top-3 | | | | |
| Real | $0.424^{\pm.005}$ | $0.649^{\pm.006}$ | $0.779^{\pm.006}$ | $0.031^{\pm.004}$ | $2.788^{\pm.012}$ | $11.08^{\pm.097}$ | - |
| MDM | $0.164^{\pm.004}$ | $0.291^{\pm.004}$ | $0.396^{\pm.004}$ | $0.497^{\pm.021}$ | $9.191^{\pm.022}$ | $10.85^{\pm.109}$ | $1.907^{\pm.214}$ |
| MotionDiffuse | $0.417^{\pm.004}$ | $0.621^{\pm.004}$ | $0.739^{\pm.004}$ | $1.954^{\pm.062}$ | $2.958^{\pm.005}$ | $\mathbf{11.10}^{\pm.143}$ | $0.730^{\pm.013}$ |
| MLD | $0.390^{\pm.008}$ | $0.609^{\pm.008}$ | $0.734^{\pm.007}$ | $0.404^{\pm.027}$ | $3.204^{\pm.027}$ | $10.80^{\pm.117}$ | $2.192^{\pm.071}$ |
| GraphMotion | $0.429^{\pm.007}$ | $0.648^{\pm.006}$ | $0.769^{\pm.006}$ | $0.313^{\pm.013}$ | $3.076^{\pm.022}$ | $\underline{11.12}^{\pm.135}$ | $\mathbf{3.627}^{\pm.113}$ |
| ReMoDiffuse | $0.427^{\pm.014}$ | $0.641^{\pm.004}$ | $0.765^{\pm.055}$ | $\mathbf{0.155}^{\pm.006}$ | $2.814^{\pm.012}$ | $10.80^{\pm.105}$ | $1.239^{\pm.028}$ |
| T2M-GPT | $0.416^{\pm.006}$ | $0.627^{\pm.006}$ | $0.745^{\pm.006}$ | $0.514^{\pm.029}$ | $3.007^{\pm.023}$ | $10.921^{\pm.108}$ | $1.570^{\pm.0.39}$ |
| AttT2M | $0.413^{\pm.006}$ | $0.632^{\pm.006}$ | $0.751^{\pm.006}$ | $0.870^{\pm.039}$ | $3.039^{\pm.021}$ | $10.96^{\pm.043}$ | $\underline{2.281}^{\pm.047}$ |
| MoMask | $0.433^{\pm.007}$ | $0.656^{\pm.005}$ | $0.781^{\pm.005}$ | $\underline{0.204}^{\pm.011}$ | $2.779^{\pm.022}$ | - | $1.131^{\pm.043}$ |
| Our (NFE 1) | $\underline{0.441}^{\pm.006}$ | $\underline{0.667}^{\pm.005}$ | $\underline{0.792}^{\pm.006}$ | $0.389^{\pm.012}$ | $2.764^{\pm.017}$ | $11.197^{\pm.102}$ | $1.562^{\pm.035}$ |
| Our (NFE 2) | $\mathbf{0.445}^{\pm.005}$ | $\mathbf{0.669}^{\pm.006}$ | $\mathbf{0.797}^{\pm.004}$ | $0.389^{\pm.016}$ | $\underline{2.740}^{\pm.020}$ | $11.216^{\pm.100}$ | $1.517^{\pm.030}$ |
| Our (NFE 4) | $\underline{0.441}^{\pm.006}$ | $0.665^{\pm.005}$ | $0.790^{\pm.007}$ | $0.343^{\pm.011}$ | $\mathbf{2.739}^{\pm.015}$ | $11.134^{\pm.098}$ | $1.552^{\pm.036}$ |

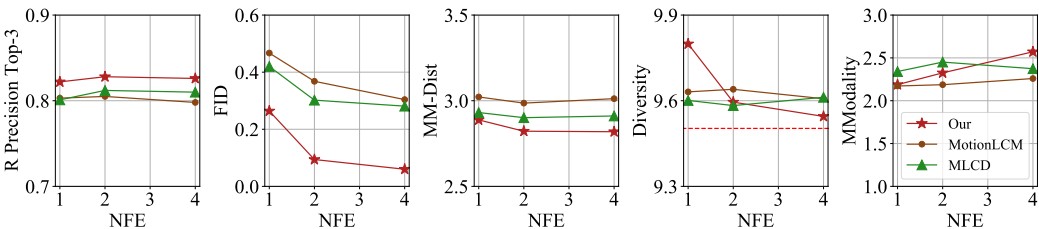

Figure 3: Comparison with latent consistency distillation frameworks, including the latest proposed MotionLCM and ablation experiments of the proposed method in distillation mode.

quantitative test results for the HumanML and KIT datasets are provided in Tables 2 and 3, respectively. The results are categorized into three areas: previous diffusion frameworks, other generative frameworks, and our proposed framework. Consistent with prior research (Tevet et al., 2023; Chen et al., 2023), we conducted all evaluations 20 times and reported the averages with a 95% confidence interval. Our approach performs comparably to state-of-the-art models. Specifically, for the HumanML3D dataset, our method surpasses previous state-of-the-art motion diffusion frameworks (Zhang et al., 2023b) across various metrics, particularly in maintaining high diversity despite increased controllability, while achieving a reduction in inference costs exceeding 70%. Moreover, our single-step inference performance is competitive, surpassing the baseline model of motion latent diffusion methods (Chen et al., 2023). For recent advances (Guo et al., 2024) in masked transformer models, our approach achieves matching performance on *FID* metric (0.060 vs. 0.045 for MoMask) with a 50% reduction in inference cost, while showing significant advantages in terms of controllability and diversity. For the KIT dataset, our method maintains the optimal controllability performance but is limited in the *FID* metric. This limitation arises from the encoding method based on quantization constraints is more sensitive to datasets with smaller sample sizes, resulting in reduced motion encoding performance. The *MultiModality* metric also exhibits challenges with small datasets but achieves performance comparable to the optimal metric for the *Diversity* metric (11.134 for our method vs. 11.10 for MotionDiffuse).

## 5.3 COMPARISONS TO CONSISTENCY DISTILLATION

We are motivated to enhance latent consistency training to achieve performance that matches or exceeds traditional latent consistency distillation. To this end, we compare our approach with our concurrent work, MotionLCM (Dai et al., 2024), which adheres to the consistency distillation framework. The test results are presented in Figure 3. Our approach consistently outperforms MotionLCM in terms of controllability, generation quality, and diversity under the same NFE. It is worth noting that MotionLCM employs pelvic control, i.e., it requires previous awareness of the real pelvic trajectory bootstrap, even during testing and inference. Considering the differences in detail between the

Table 4: Ablation study about each part of our method on the HumanML3D test set. Marker meaning is consistent with Table 2.

| Quantization Constraint | Conditionally Guided CT | Clustering Guidance | R-Precision Top-3 ↑ | FID ↓ | MM-Dist ↓ | Diversity → (9.503) | MModality ↑ |
|---|---|---|---|---|---|---|---|
| ✗ | ✗ | ✗ | $0.639^{\pm.006}$ | $2.651^{\pm.021}$ | $4.021^{\pm.103}$ | $8.421^{\pm.040}$ | $3.909^{\pm.040}$ |
| ✓ | ✗ | ✗ | $0.734^{\pm.004}$ | $0.615^{\pm.006}$ | $3.351^{\pm.008}$ | $9.248^{\pm.084}$ | $3.961^{\pm.059}$ |
| ✗ | ✓ | ✗ | $0.778^{\pm.005}$ | $0.541^{\pm.008}$ | $3.201^{\pm.008}$ | $9.012^{\pm.093}$ | $2.570^{\pm.042}$ |
| ✗ | ✗ | ✓ | $0.634^{\pm.004}$ | $2.596^{\pm.010}$ | $4.036^{\pm.007}$ | $9.401^{\pm.086}$ | $4.063^{\pm.065}$ |
| ✓ | ✓ | ✗ | $0.821^{\pm.002}$ | $0.210^{\pm.005}$ | $2.886^{\pm.009}$ | $9.535^{\pm.069}$ | $2.411^{\pm.050}$ |
| ✓ | ✗ | ✓ | $0.733^{\pm.004}$ | $0.542^{\pm.005}$ | $3.351^{\pm.007}$ | $9.234^{\pm.055}$ | $4.079^{\pm.043}$ |
| ✗ | ✓ | ✓ | $0.784^{\pm.003}$ | $0.454^{\pm.007}$ | $3.017^{\pm.008}$ | $9.137^{\pm.034}$ | $2.346^{\pm.078}$ |
| ✓ | ✓ | ✓ | $0.826^{\pm.002}$ | $0.060^{\pm.003}$ | $2.819^{\pm.010}$ | $9.545^{\pm.068}$ | $2.571^{\pm.051}$ |

Table 5: Ablation study of different token counts $n$ on the HumanML3D test set. Marker meaning is consistent with Table 2.

| $n$ | R-Precision Top-3↑ | FID ↓ | MModality ↑ |
|---|---|---|---|
| 1 | $0.810^{\pm.002}$ | $0.249^{\pm.009}$ | $2.935^{\pm.067}$ |
| 2 | $0.804^{\pm.003}$ | $0.210^{\pm.005}$ | $2.872^{\pm.069}$ |
| 3 | $0.814^{\pm.003}$ | $0.136^{\pm.005}$ | $2.828^{\pm.068}$ |
| 4 | $0.826^{\pm.002}$ | $0.060^{\pm.003}$ | $2.571^{\pm.051}$ |
| 5 | $0.826^{\pm.002}$ | $0.094^{\pm.010}$ | $2.716^{\pm.065}$ |

Table 6: Ablation study of different quantization levels $l$ on the HumanML3D test set. Marker meaning is consistent with Table 2.

| $l$ | R-Precision Top-3 ↑ | FID ↓ | MModality ↑ |
|---|---|---|---|
| 128 | $0.814^{\pm.002}$ | $0.113^{\pm.004}$ | $2.612^{\pm.065}$ |
| 256 | $0.826^{\pm.002}$ | $0.060^{\pm.003}$ | $2.571^{\pm.051}$ |
| 512 | $0.825^{\pm.003}$ | $0.121^{\pm.005}$ | $2.721^{\pm.043}$ |
| 1024 | $0.812^{\pm.003}$ | $0.142^{\pm.005}$ | $2.872^{\pm.072}$ |
| 2048 | $0.819^{\pm.002}$ | $0.134^{\pm.007}$ | $2.848^{\pm.066}$ |

Table 7: Ablation study of different guidance scales $\omega$ on the HumanML3D test set. Marker meaning is consistent with Table 2.

| $\omega$ | R-Precision Top-3↑ | FID ↓ | MModality ↑ |
|---|---|---|---|
| 1 | $0.806^{\pm.002}$ | $0.250^{\pm.008}$ | $2.958^{\pm.068}$ |
| 2 | $0.813^{\pm.002}$ | $0.213^{\pm.007}$ | $2.689^{\pm.046}$ |
| 3 | $0.820^{\pm.003}$ | $0.145^{\pm.004}$ | $2.532^{\pm.064}$ |
| 4 | $0.826^{\pm.002}$ | $0.060^{\pm.003}$ | $2.571^{\pm.051}$ |
| 5 | $0.825^{\pm.002}$ | $0.101^{\pm.009}$ | $2.442^{\pm.066}$ |

Table 8: Ablation study of different clustering counts $k$ on the HumanML3D test set. Marker meaning is consistent with Table 2.

| $k$ | R-Precision Top-3 ↑ | FID ↓ | MModality ↑ |
|---|---|---|---|
| 256 | $0.823^{\pm.002}$ | $0.129^{\pm.003}$ | $2.537^{\pm.059}$ |
| 512 | $0.825^{\pm.002}$ | $0.130^{\pm.004}$ | $2.545^{\pm.064}$ |
| 1024 | $0.823^{\pm.003}$ | $0.113^{\pm.004}$ | $2.567^{\pm.074}$ |
| 2048 | $0.826^{\pm.002}$ | $0.060^{\pm.003}$ | $2.571^{\pm.051}$ |
| 4096 | $0.829^{\pm.002}$ | $0.098^{\pm.003}$ | $2.549^{\pm.071}$ |

two approaches, we implemented latent consistency distillation with quantized representation and clustering guidance, referred to as MLCD, with results also depicted in Figure 3. The experiments demonstrate that our proposed enhancement techniques contribute to the consistency distillation. The advantage of consistency training lies in its independence from the performance of the pre-trained model, allowing it to exhibit greater potential. Additionally, it avoids the costs associated with pre-training, reducing both computational and time overhead in the training process.

## 5.4 ABLATION STUDY

**Effectiveness of each component.** To further investigate the contributions of the proposed technique, we conducted ablation experiments for each combination of components within the approach and presented the results in Table 5. For clarity, when the quantization constraint is denoted as ✗, it signifies that the model utilizes the variational autoencoder with KL constraints; conversely, our proposed quantized encoder incorporates quantization constraints. The experiments indicate that the performance of consistency training significantly declines in the absence of any optimization techniques. Each proposed boosting scheme enhanced the results to varying degrees, with conditionally guided contributions yielding the highest improvements. The integration of the three techniques achieved state-of-the-art performance, underscoring the effectiveness of the proposed method.

**Ablation study on the different model hyperparameters.** For motion encoding, we present ablation experiments with different token counts $n$ and quantization levels $l$ in Table 5 and 6. Unlike MLD where more tokens are less effective, increasing the token counts in our framework significantly improves the generation quality. Experimentally, lower quantization levels $l$ result in a more

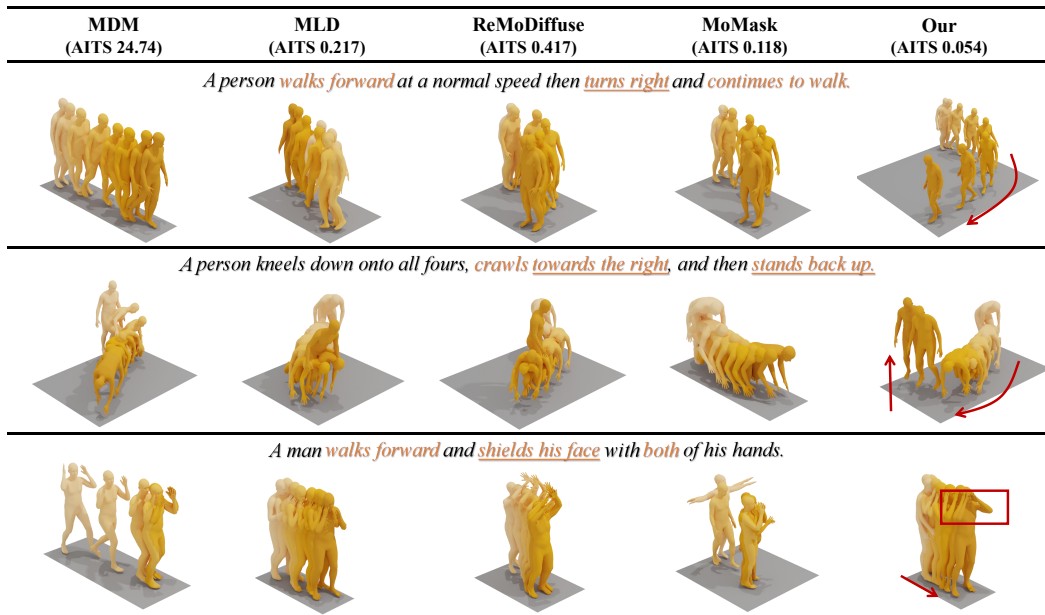

| MDM
(AITS 24.74) | MLD
(AITS 0.217) | ReMoDiffuse
(AITS 0.417) | MoMask
(AITS 0.118) | Our
(AITS 0.054) |
|---|---|---|---|---|

*A person walks forward at a normal speed then turns right and continues to walk.*

*A person kneels down onto all fours, crawls towards the right, and then stands back up.*

*A man walks forward and shields his face with both of his hands.*

Figure 4: Qualitative analysis of our model and previous models. We present three texts to guide the motion visualization results. Our model demonstrates improved motion generation performance, matching textual conditions with lower inference costs. The color of humans darkens over time.

concise solution space but impact the reconstruction performance of the decoder. At larger token counts, the conciseness from lower quantization levels is traded off with the reconstruction performance. For the guidance scale $\omega$, we demonstrate the test results in Table 7. We observed that various levels of the guidance scale contribute positively to generation quality. As the guidance scale $\omega$ increases, controllability gradually improves, with a corresponding decrease in diversity. This is consistent with previous experience with CFG techniques in diffusion inference. For the number of clustering categories, we show the ablation experiment results in Table 8. The experiments show no contribution to generation performance when the clustering category number is small, whereas a larger number allows for more fine-grained guidance.

## 5.5 TIME COST AND QUALITATIVE RESULTS

We present a qualitative analysis of our approach with two baselines (MDM and MLD) and two state-of-the-art models (ReMoDiffuse and MoMask) in Figure 4. While previous works have accurately captured the general instruction semantics, they insufficiently responded to the details such as orientation. In contrast, our approach enables the generation of fine-grained, high-quality motions with reduced inference time.

## 6 CONCLUSION

In this paper, we present a motion latent consistency training framework, designed for fast, high-fidelity, text-matched motion generation. This framework encodes human motion sequences into tokens using a quantization constraint, which ensures bounded finite states to optimize the latent representation. Additionally, we propose a conditionally guided consistency training framework and a clustering guidance module to enhance conditional controllability and provide supplementary solution distribution references. Our model and its components have been validated through extensive experiments, demonstrating an optimal trade-off between performance and computational efficiency with minimal NFE. Our approach serves as a reference for training subsequent latent consistency training across various tasks.

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

This appendix provides additional discussions (Section A), more implementation details (Section B), more qualitative results (Section C), user study (Section D), and details of evaluation metric (Section E).

During the review phase, our code is available in the anonymous repository [1].

# A  ADDITIONAL DISCUSSIONS

## A.1  INFERENCE COSTS OF COMPONENTS

We present the overall inference costs in Table 1. To aid readers in evaluating the efficiency of each component, we measured the average single inference costs: the text encoder is 0.0186 seconds, the clustering guidance module is 0.0008 seconds, the denoiser is 0.0071 seconds, and the motion decoder is 0.0042 seconds. Notably, text encoding is relatively time-consuming, and the time cost associated with clustering guidance during the inference process is minimal.

## A.2  POTENTIAL NEGATIVE SOCIETAL IMPACTS

Our work enhances the efficiency of human motion synthesis and may be applied to generate fake information, which may threaten information security and intellectual property rights. In embodied intelligence, it may generate irrational robot joint mappings, which may cause property damage and security risks.

## A.3  LIMITATION

Our work still has some directions for improvement: (i) The MLCT follows the diffusion modeling framework, and its stochastic nature favors diversity, but may sometimes produce undesired results. Additionally, our frameworks learn distributions directly from data without involving physical laws. This concern also arises in previous work such as GraphMotion or MLD. (ii) Improving the robustness of latent autoencoders on small datasets is an open question. The performance of the proposed framework on small datasets needs to be further explored. (iii) Our set of textual instructions focuses on the annotated data of HumanML3D, but it may be limited, and out-of-domain instructions may occur resulting in unreasonable sample generation.

## A.4  FUTURE WORK

We would like to include more physical constraints in our follow-up work to minimize undesired motion generation and adopt a more appropriate text extractor for fine-grained motion control. Noting the rise of large language models, subsequent works could utilize them to assist in understanding a broader context of semantic instructions. In addition, zero-shot editing for consistency training based on large language models is also worthy of research.

# B  MORE IMPLEMENTATION DETAILS

For balance training, we set $\lambda_j$ as $10^{-3}$. Following the ablation experiments, we set the guidance scale $\omega$ to 4. All the proposed models are trained with the AdamW optimizer with a learning rate of $10^{-4}$. For diffusion time horizon $[\epsilon, T]$ into $N-1$ sub-intervals, we set $\epsilon$ is 0.002, $T$ is 1, $N$ is 25. We follow the consistency model (Song et al., 2023) to determine $t_i = (\epsilon^{1/\rho} + \frac{i-1}{N-1}(T^{1/\rho} - \epsilon^{1/\rho}))^\rho$, where $\rho = 7$. In addition, we set the EMA rate to $\gamma = 0.995$ in all experiments. For better reproducibility, we provide pseudo-code for training and inference, as shown in Algorithm 1 and Algorithm 2, respectively.

---

[1] https://anonymous.4open.science/r/Efficient-Text-driven-Motion-Generation-via-Latent-Consistency-Training-E4EF

---

**Algorithm 1:** Motion Latent Consistency Training.

---

**Input:** Train set $\Gamma = \{(x^{(n)}, c^{(n)})\}_{n=1}^{N}$, Motion AutoEncoder $\mathcal{G} = \{\mathcal{E}, \mathcal{D}\}$ with initial parameter $\theta$, size $2l + 1$ of finite set $\mathcal{M}$, Joint Transform Function $\mathcal{J}$, Motion Consistency Model $\mathcal{S}$ with initial parameter $\psi$ and $\psi^{-}$, ODE Solver $\Phi$, Timestep Scheduler $\{t_i\}_{i=0}^{I}$, Guidance Scale $\omega$, Learning Ratio $\eta$, EMA Ratio $\gamma$, Balance Weight $\lambda_j$;

1 **# Stage 1: Motion AutoEncoder Training.**
2 **repeat**
3     Sample motion $x \sim \Gamma$;
4     $z_e \leftarrow \mathcal{E}(x)$;                  // Motion Encoding.
5     $z_m \leftarrow \mathcal{R}\Big(l \cdot tanh(z_e)\Big)/l$;        // Quantization Constraint.
6     $\mathcal{L}_z \leftarrow \mathbb{E}_x\Big[d\big(x, \mathcal{D}(z_m)\big) + \lambda_j d\big(\mathcal{J}(x), \mathcal{J}(\mathcal{D}(z_m))\big)\Big]$;       // Loss.
7     $\theta \leftarrow \theta - \eta \nabla_{\theta} \mathcal{L}_z$.               // Update $\theta$.
8 **until** convergence
9 **# Stage 2: Motion Consistency Training.**
10 **repeat**
11     Sample motion $x$ and condition $c \sim \Gamma$, noise $z \sim \mathcal{N}(0, I)$, timestep $t_i, t_{i-1} \sim \{t_i\}_{i=0}^{I}$;
12     $x_{\epsilon} \leftarrow \mathcal{R}\Big(l \cdot tanh(\mathcal{E}(x))\Big)/l$;          // Motion Encoding.
13     $x_{t_i} \leftarrow \alpha_{t_i} \cdot x_{\epsilon} + \sigma_{t_i} \cdot z$; // Perturbed Data. $\alpha_t$ and $\sigma_t$ Detailed in Equ. 10.
14     $x_{\epsilon}^{\Phi} \leftarrow (1 + \omega) \cdot x_{\epsilon} - \omega \mathcal{S}_{\psi}(x_{t_i}, t_i, \emptyset)$;      // CFG in Consistency Training.
15     $x_{\epsilon}^{\Phi} \leftarrow \text{clamp}(x_{\epsilon}^{\Phi}, -1, 1)$;             // Clamp.
16     $\widetilde{x}_{t_{i-1}} \leftarrow \Phi(x_{\epsilon}^{\Phi}, t_i, t_{i-1})$;     // One-step Numerical Estimation with Equ. 12.
17     $\mathcal{L}_c \leftarrow \mathbb{E}_{x,t}\Big[\frac{1}{t_i - t_{i-1}} d\big(\mathcal{S}_{\psi}(x_{t_i}, t_i, c), \mathcal{S}_{\psi^-}(\widetilde{x}_{t_{i-1}}, t_{i-1}, c)\big) + d\big(\mathcal{S}_{\psi}(x_{t_i}, t_i, \emptyset), x_{\epsilon}\big)\Big]$;    // Loss.
18     $\psi \leftarrow \psi - \eta \nabla_{\psi} \mathcal{L}_c$;             // Update $\psi$.
19     $\psi^{-} \leftarrow \text{stopgrad}(\gamma \psi^{-} + (1 - \gamma)\psi)$.       // Update $\psi^{-}$.
20 **until** convergence

---

**Algorithm 2:** Motion Latent Consistency Inferring.

---

**Input:** Motion AutoEncoder $\mathcal{G} = \{\mathcal{E}, \mathcal{D}\}$, Joint Transform Function $\mathcal{J}$, Motion Consistency Model $\mathcal{S}$, Condition $c$, Max Number of Function Evaluations $N$, Timestep Scheduler $\{t_i\}_{i=0}^{N}$;

**Result:** Motion Sequence $x$.

1 Sample $x_{t_N}, z \sim \mathcal{N}(0, I)$;
2 **for** i=N to 1 **do**
3     **if** i != N **then**
4        $x_{t_i} \leftarrow \alpha_{t_i} \cdot x_{\epsilon} + \sigma_{t_i} \cdot z$;             // Perturbed Data.
5     $x_{\epsilon}^{pred} \leftarrow \mathcal{S}_{\psi}(x_{t_i}, t_i, c)$;             // Denoising.
6     $x_{\epsilon}^{pred} \leftarrow \text{clamp}(x_{\epsilon}^{pred}, -1, 1)$;          // Clamp.
7 $x = \mathcal{D}(x_t)$.                   // Motion Decoding.

---

## C   More Qualitative Result

We show the more qualitative results under the few NFE in Figure 5.

## D   User Study

Following the configuration in MLD, we set up UserStudy. We randomly generated 30 sets of text descriptions in the test set of the HumanML3D dataset and used MLCT and MLD to generate the corresponding text, respectively. We invited 36 participants to provide two comparisons: the MLCT and the MLD, and the MLCT and the ground truth motion in the dataset. Each set of motions will be compared for fidelity and condition matching. The results are reported in Figure 6. Our method outperforms MLD with a low inference cost of 4 NFE and is even competitive with ground truth results provided by motion capture devices in terms of fidelity and condition matching.

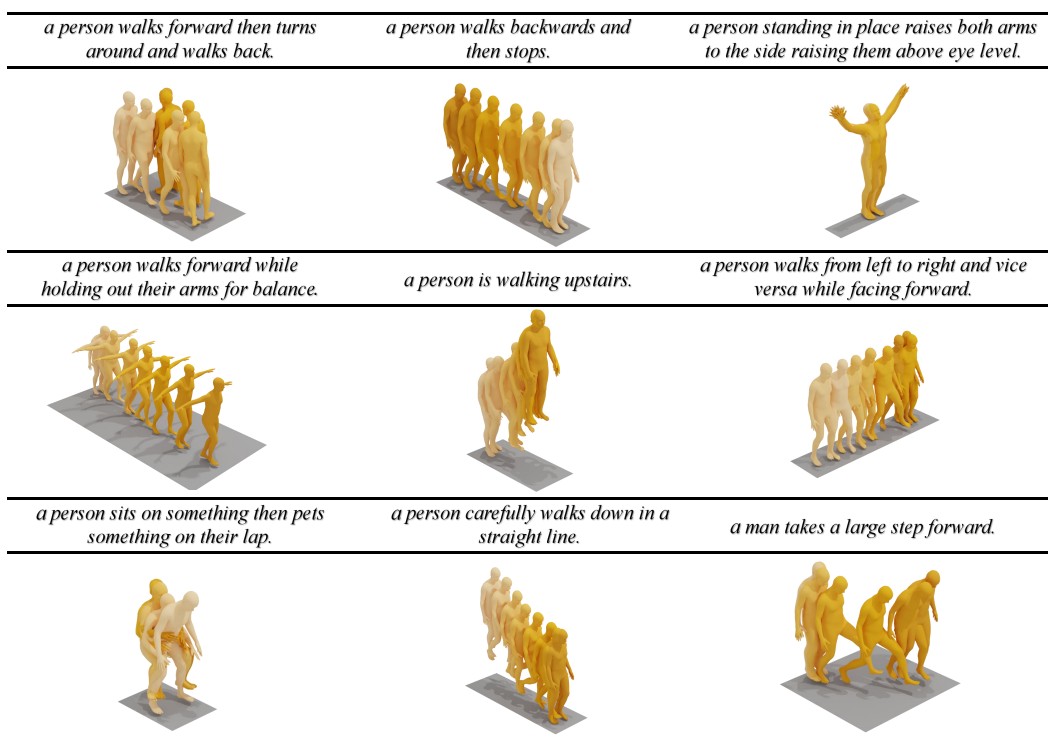

Figure 5: More samples from our model for text-to-motion synthesis, which was trained on the HumanML3D dataset. The color of humans darkens over time.

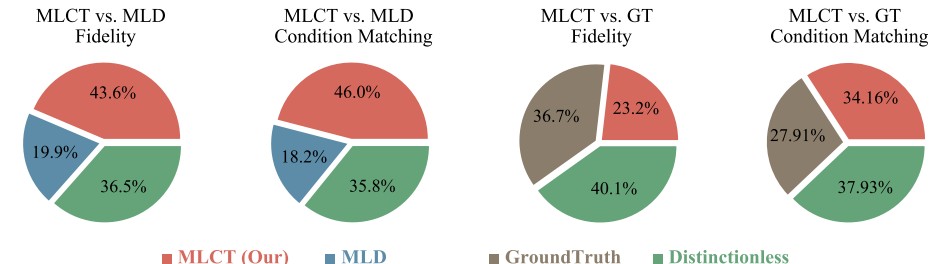

Figure 6: User studies for quantitative comparison. We follow MLD Chen et al. (2023) that utilizes the force-choice paradigm to ask "which of the two motions is more realistic?" and "which of the two motions corresponds better to the text prompt?" We show the preference rate of MLCT over the MLD and Ground Truth data.

## E    DETAILS OF EVALUATION METRIC

We utilize the standard feature extractor (Guo et al., 2022) to calculate the features of motions and texts. The parameters of metrics are consistent with previous work (Chen et al., 2023; Jin et al., 2023).

**Frechet Inception Distance (FID).** FID is the principal metric to evaluate the generation quality, which examines the similarity between the generated motion distribution and the ground truth motion distribution. It is formalized as:

$$\text{FID} = \|\mu_{gt} - \mu_{gen}\|^2 + \text{Tr}(\Sigma_{gt} + \Sigma_{gen} - 2(\Sigma_{gt} \cdot \Sigma_{gen})^{\frac{1}{2}}), \tag{17}$$

where $\mu$ and $\Sigma$ denote the mean and the covariance matrix of motion features, and Tr denotes the trace of the corresponding matrix.

**R-Precision.** Given a motion feature and 32 textual descriptions (one of ground truth and the others are randomly selected mismatched descriptions), we calculate the matching accuracy of text and motion for Top 1/2/3.

**Multimodal Distance (MM-Dist).** For $N$ randomly generated motions, we calculate the average Euclidean distances between motion features and text features. It is formalized as:

$$\text{MM-Dist} = \frac{1}{N} \sum_{i=1}^{N} \|f_{m,i} - f_{t,i}\|, \tag{18}$$

where $f_{m,i}$ and $f_{t,i}$ denote the feature of $i$-th motion and text.

**Diversity.** We calculate the average Euclidean distances between two randomly divided groups of generated motion features $\{x\}_{i=1}^{N}$ and $\{x'\}_{i=1}^{N}$. It is formalized as:

$$\text{Diversity} = \frac{1}{N} \sum_{i=1}^{N} \|x_i - x_i'\|. \tag{19}$$

**Multimodality (MModality).** For $J$ text descriptions, we randomly sampled two subsets of the same size $N$ from all motions generated by $j$-th text descriptions, with motion features $\{x_{j,1}, \cdots, x_{j,N}\}$ and $\{x_{j,1}', \cdots, x_{j,N}'\}$. We calculate the average Euclidean distance formalized as:

$$\text{MModality} = \frac{1}{J \times N} \sum_{j=1}^{J} \sum_{n=1}^{N} \|x_{j,n} - x_{j,n}'\|. \tag{20}$$

**Average Inference Time per Sentence (AITS).** We repeatedly test $N$ times for generating the longest motion, and report the average inference time.

