# OpenReview forum: "Efficient Text-driven Human Motion Generation via Latent Consistency Training"
_ICLR.cc/2025/Conference — ICLR 2025 Conference Withdrawn Submission_

### Official Review · Reviewer_K6N8 · 2024-10-15

**Soundness:** 3
**Presentation:** 2
**Contribution:** 2
**Rating:** 5
**Confidence:** 3

**Summary:**

Overall, considering the success and advantages of consistency training, this paper extends consistency training from the pixel space to the latent space for the first time. Meanwhile, this paper also correspondingly involves some designs such as over the training-time CFG.

**Strengths:**

1. Based on the importance of consistency training, I believe that this paper addresses an important problem of extending consistency training to the latent space.

2. The paper does achieve a quite efficient inference as claimed.

**Weaknesses:**

(see the questions part for more details.)

**Questions:**

Overall, I believe that this paper currently is roughly on the borderline level. Below are my concerns.

1. Fisrtly, I believe that Figure 1 needs to be adjusted to also involve "traditional" consistency training as (c) and the proposed latent consistency training as (d). This is to more clearly compare the proposed method and the existing consistency training pipeline. The current drawing can confuse the readers on the advantage of consistency training and the advantage of the proposed method itself.

2. I suggest the authors to more explicitly point out the challenge of extending consistency learning from the pixel space to the latent space. The current manuscript seems to focus more on how this is done in the proposed method but not why this is challenging, leading to difficulty in better understanding the contribution of this work.

3. I believe that the sensitivity of the quantization level l across different datasets is important to be measured. Specifically, I am curious that whether the quantization level would cause different level of information loss among different datasets.

4. W.r.t Sec. 4.3, the authors seem to cluster text into different clusters and provide motion clustering guidance. I am wondering if this could lead the effectiveness of the guidance, for text near and far from the center of the cluster, to be different.

5. W.r.t. table 1 and 2, I also have the following suggestions:

(1) It is suggested for the model training time for those methods that need to train the parent model (e.g., distillation-based methods) to be also reported to better show the advantage of the proposed method.

(2) It is suggested to combine table 1 and 2 to give readers a better feeling over the performance vs the efficiency.

---

### Official Review · Reviewer_nXgd · 2024-10-29

**Soundness:** 3
**Presentation:** 2
**Contribution:** 2
**Rating:** 5
**Confidence:** 4

**Summary:**

This paper presents a motion diffusion model with consistency regularization in latent space. This paper encode the motion into a latent space with quantization. They also simulate the conditional generation and design a similar loss for training. They also introduce a cluster center guidance to map text center to motion center.

**Strengths:**

1. This paper shows low inference cost for motion synthesis, and is competitive with the prevailing MotionLCM.
2. This paper presents good results on KIT and HumanML3D, and conducts many ablation study.

**Weaknesses:**

1. The quantization of latent code is widely used and the cross-attention for text and motion is common. These two modules are not highly related to the consistency training. This can also be reflected in Figure 3.
2. In Table 2 and Table 3, the motions synthesized by the proposed method show results than the real data in Top-k R-precision, and this should be explained.
3. According to Table 3, the Modality is sacrificed during the consistency training.
4. Some contents are not self-contained. For example, in Eq. 4, why \mathcal{S}=\epsilon is not explained. In Eq. 5, the \hat{\mathcal{S}} is not explained. In line 266, x_{\epsilon} and x_0 can be confusing.

**Questions:**

The definition of each symbol should be given clearly for easy read.

---

### Official Review · Reviewer_tPxo · 2024-11-04

**Soundness:** 3
**Presentation:** 2
**Contribution:** 2
**Rating:** 5
**Confidence:** 5

**Summary:**

This paper proposes a novel framework for efficient text-to-motion synthesis, which requires one third of inference time compared to the existing state-of-the art motion synthesis work, MoMask. Different from existing latent consistency model, e.g., MotionLCM, the current framework gains insight from consistency training in pixel-based space, and introduce a bounded quantization method which converts continuous motions to the discrete value ranging from -1 to 1 like pixels. Meanwhile, consistency constraints are involved in both training and inference, potentially go beyond the pre-trained diffusion model. Quantitative evaluations showed that the proposed method performed comparable to existing state-of-the-art methods while being faster two time than MoMask, and being the same fast as latests motion consistency model.

**Strengths:**

**1. Technically novel**. This work proposes a novel scheme of motion latent consistency models. I like the idea of how the author tailored the motion representation as a pixel-like representation using the bounded quantization. Different from existing motion consistency model, MotionLCM, which tried to distill knowledge from pre-trained diffusion model and thus is limited to the performance of these diffusion models,  the current work incorporate consistency constraints in both training and inference. Meanwhile, other novel techniques such as CFG in consistency training to enhance the semantic matchness. Overall, this work proposes a fundamentally different framework for efficient text-to-motion synthesis. However, since my expertise is not on the theoretical side, I may not be able to validate the theories described in the paper.
**2. Comprehensive evaluations**. I appreciate the authors effort in conducting a series of experiments for baseline comparisons, ablation analysis. From table 4-8, we can observe the effect of token number, quantization level, and clustering counts.
**3. Outperforming quantitative evaluations**. From table 2-3, the proposed method achieves comparable performance (i.e., FID, RPrecision) comparing to non-consistency model, which being two times faster. Compared to existing motion consistency mode, i.e., MotionLCM, the proposed method costs the same time will performs better on motion quality and semantic matchness.

**Weaknesses:**

**1. Presentation need improvement**. The presentation is kind of messy in this paper. Many symbols in Figure 1 & 2 are corrupted, which makes them hard to understand. Without this two figures, it's troublesome for the readers to understand the main logic and points in the proposed framework. Though there are detailed descriptions in text, understanding the whole picture is still challenging, especially for connecting all the pieces together.
**2. Qualitative results look mediocre**. Although the quantitative results show the effectiveness of the proposed consistency model, while in the visualizations the difference are rather subtle. BTW, I think it's necessary to visually compare to motionLCM as well, as it's the most related work. In addition, from the visual results, we can observe several artifacts such as foot sliding, motionless movement, etc. It also seemed that the authors use different length for MoMask and other diffusion based method, which may not be fair.
**3. Limited significance**. The current work aims to address the efficiency of text-to-motion that, however, not seems not an important problem any more. Unlike images/videos, motions are low-dimensional data. After the work of MLD & MoMask, the time cost for motion synthesis is not that unacceptable. Though this work proposes a more lightweight inference scheme, the time cost is not substantially different, only from 0.118 to 0.031. For me, the significance of this work is questionable.

**Questions:**

Please refer to the weakness section. The presentation and figures are yet to be ready. I also hope the authors could properly justify the significancy of this work.

---

### Note · Authors · 2024-11-26

**Comment:**

I would like to sincerely thank the ICLR organizers and reviewers for their valuable feedback and support. Due to unforeseen circumstances and the need to further refine my work, I have decided to withdraw this submission. I deeply appreciate the time and effort invested in reviewing my paper and hope to contribute again in the future.

**Withdrawal Confirmation:**

I have read and agree with the venue's withdrawal policy on behalf of myself and my co-authors.